# Immunoinformatics Approach to Design Multi-Epitope- Subunit Vaccine against Bovine Ephemeral Fever Disease

**DOI:** 10.3390/vaccines9080925

**Published:** 2021-08-19

**Authors:** Shruti Pyasi, Vinita Sharma, Kumari Dipti, Nisha Amarnath Jonniya, Debasis Nayak

**Affiliations:** 1Discipline of Biosciences and Biomedical Engineering, Indian Institute of Technology Indore, Indore 452020, India; phd1501171002@iiti.ac.in (S.P.); phd1601271002@iiti.ac.in (N.A.J.); 2Department of Biochemistry, Central University of Rajasthan, Rajasthan 305817, India; vinusharma650@gmail.com; 3Department of Bioscience & Biotechnology, Banasthali Vidyapith, Rajasthan 304022, India; diptirkp255@gmail.com

**Keywords:** BEFV, Immunoinformatics, multi-epitopes vaccine, BoLA, Toll-like receptor 7

## Abstract

Bovine ephemeral fever virus (BEFV) is an overlooked pathogen, recently gaining widespread attention owing to its associated enormous economic impacts affecting the global livestock industries. High endemicity with rapid spread and morbidity greatly impacts bovine species, demanding adequate attention towards BEFV prophylaxis. Currently, a few suboptimum vaccines are prevailing, but were confined to local strains with limited protection. Therefore, we designed a highly efficacious multi-epitope vaccine candidate targeted against the geographically distributed BEFV population. By utilizing immunoinformatics technology, all structural proteins were targeted for B- and T-cell epitope prediction against the entire allele population of BoLA molecules. Prioritized epitopes were adjoined by linkers and adjuvants to effectively induce both cellular and humoral immune responses in bovine. Subsequently, the in silico construct was characterized for its physicochemical parameters, high immunogenicity, least allergenicity, and non-toxicity. The 3D modeling, refinement, and validation of ligand (vaccine construct) and receptor (bovine TLR7) then followed molecular docking and molecular dynamic simulation to validate their stable interactions. Moreover, in silico cloning of codon-optimized vaccine construct in the prokaryotic expression vector (pET28a) was explored. This is the first time HTL epitopes have been predicted using bovine datasets. We anticipate that the designed construct could be an effective prophylactic remedy for the BEF disease that may pave the way for future laboratory experiments.

## 1. Introduction

Bovine ephemeral fever virus (BEFV) is a widespread and economically significant cattle pathogen associated with bovine ephemeral fever (BEF) disease. The neglected tropical disease dates back to its first emergence in the mid-nineteenth century in Africa, then subsequently became endemic to Australia, Africa, the Middle East, and major parts of the Asiatic continent including East Asia and South-East Asia [1,2]. BEF has a high morbidity rate (up to 100%). The disease has shown enhanced pathogenicity, spreading to newer geographical regions in recent years [3,4]. BEFV outbreaks are perceptible during monsoon onset, coinciding with vectors’ propagation (*Culicoides* biting midges and mosquitoes) and spreading the disease [5,6]. The disease symptoms include polyphasic fever, synovitis, somnolence, muscle stiffness, lameness, reluctance to move, inappetence, paralysis, and ataxia (of hind limbs). [7]. Severely affected cows show a significant reduction in quality and quantity of milk production, often develop milk withdrawal, and in some cases abort the fetus in the third trimester. Affected bulls show lower infertility and poor breeding value. Moreover, the virus predominantly targets crossbred and healthy cattle and water buffaloes [8]. Altogether, BEFV cause heavy loss to cattle farming in multiple fronts including milk, meat, traction losses at the field, and restriction of international livestock trade.

BEFV belongs to *Rhabdoviridae* family and *Ephemerovirus* genus [9]. The virion contains a ~14.9 kb long, non-segmented, negative-sense single-stranded (NSSS) RNA genome that encodes five structural proteins (N, P, M, G, L) and five non-structural proteins (G_NS_-α1-α2-β-γ) [10]. Among the structural ones, glycoprotein (G) plays a crucial role in the attachment, entry, and release of viruses. It contains a highly conserved immunodominant epitope [11] and stimulates the production of host-neutralizing antibodies, and is regarded as one of the major antigenic determinants [12]. Beneath the glycoprotein envelope lies the matrix protein (M), responsible for virion packaging and budding. The nucleoprotein (N), in association with genomic RNA, phosphoprotein (P), and RNA-dependent RNA polymerase (L) protein, forms a biologically active nucleocapsid core (NC). The NC, which promotes the formation of the transcription-replication complex, is central to viral propagation [10]. The N protein is often immunogenic and induces a proliferative T-cell response in affected cattle [13]. Compared with structural proteins, the functions of non-structural proteins are yet to be fully explored and, therefore, are not included in our study design [14].

Only recently, the awareness of BEFV prophylaxis gained attention. Although a few BEFV vaccines are now available commercially, they revealed only limited effectiveness [15]. A few inactivated vaccines showed high neutralizing antibody titers to BEFV challenge, but failed to trigger effective cell-mediated immune (CMI) responses [16]. This necessitates the designing of candidate vaccines with better CMI response in addition to the induction of high quality neutralizing antibody titer. While recombinant vaccines utilizing various viral vector platforms have been employed, they may impose excessive antigenicity and undue risk of allergenicity [16,17]. However, a few subunit vaccines based on antigenic G protein seemed potent, but were locally confined and demanded timely booster doses, while others required clinical exploration to prove their potency [18]. Overall, the lack of effective therapeutics urges a need to develop a potent vaccine that targets all global BEFV strains with higher efficacy.

Recently, with the advancement in immunoinformatics technology, coupled with the understanding of host immune response, a new discipline for designing potent multi-epitope vaccines (MEVs) has emerged, which robustly boosted the vaccine developmental process [19]. An efficient MEV should comprise potential antigenic epitopes derived from viral proteins along with adjuvants to induce an optimal protective immune response with increased efficiency [20]. The epitopes must possess HLA binding motifs to host major histocompatibility complex (MHC) molecules. However, in cattle, the MHCs are known as bovine leukocyte antigen (BoLA) molecules, equivalent to mammalian MHC in structure and functions [21]. As virus-derived epitopes mimic natural pathogens, they hold the potential to trigger humoral and CMI responses with a minimized risk of allergenic reactions [22,23]. The aforementioned strategy suffices its potency to target various pathogens, including humans and livestock [24,25,26].

The present study utilized a range of immunoinformatics tools to design the first MEV against BEFV for efficient protection against this pathogen. We identified highly antigenic epitopes (B-cell, cytotoxic T-lymphocyte (CTL), and helper T-lymphocyte (HTL) epitopes) of viral proteins and explored for sequence conservancy among all BEFV isolates. The current vaccine design comprises all prioritized epitopes, conjugated using suitable linkers and adjuvants for optimal immune response. Using the computational biology approach, we thoroughly evaluated various immunological and physicochemical parameters, such as stability, flexibility, and solubility. A 3D model of the construct was built, refined, quality assessed, and validated.

Further, the binding affinity of the construct towards the bovine Toll-like receptors-7 (*b*TLR7) was evaluated by molecular docking and was further ascertained by molecular dynamics (MD) simulation to confirm its stability and associated interactions. Lastly, in silico, cloning of the final construct with codon-optimization was done in a prokaryotic expression system for future large-scale production with enhanced translation efficiency. The graphical description of the BEFV transmission and workflow used for designing a multi-epitope vaccine against the virus is shown in Figure 1.

## 2. Materials and Methods

### 2.1. Genome Retrieval and Protein Curation 

The complete proteome of all eight globally available BEFV sequences was retrieved in FASTA format from the NCBI database (https://www.ncbi.nlm.nih.gov, accessed on 19 December 2020). The study design comprised all the structural proteins (N, P, M, and G), except L. The protein antigenicity was checked using VaxiJen 2.0 (http://www.ddg-pharmfac.net/vaxijen/VaxiJen/VaxiJen.html, accessed on 19 December 2020) [27], at a set threshold of 0.4 that followed for epitope prediction.

### 2.2. Epitope Prediction

#### 2.2.1. Cytotoxic T-Lymphocytes (CTL) Epitope Prediction

The antigen presentation by MHC-I to the CTL is the preliminary step to trigger an immune response against viral diseases [28,29]. For the prediction of CTL-epitopes specific to BoLA molecules, we utilized the NetMHCpan 4.1 server (http://www.cbs.dtu.dk/services/NetMHCpan/, accessed on 24 December 2020) [30], as it covers binding affinity data of numerous bovine alleles [30]. The server utilizes artificial neural networks to predict a peptide’s affinity to bind to any MHC-I molecules of a known sequence, as determined by the highest prediction score and % rank <0.5 as the set threshold to predict epitopes. The server was run against all the available sets of BoLA alleles (see Appendix A) over the invariant proteome datasets.

#### 2.2.2. Helper T-Lymphocytes (HTL) Epitope Prediction

The HTL contributes a significant role in stimulating both cellular and humoral immune responses as they recognize MHC-II peptides obtained from extraneous protein derived from the extracellular environment. The HTL epitopes, therefore, present a valuable role in designing immunotherapeutic vaccines; hence, they were predicted using NetMHCIIpan 2.1 server (http://www.cbs.dtu.dk/services/NetMHCIIpan-2.1/, accessed on 2 January 2021) [31], which showed a high affinity for T_H_-cell activation. The server is based on the advanced NN-align method that utilizes two steps to estimate the network weight configuration and peptide binding score (core). It works to provide the environment for peptide binding strength with the pseudo-MHC sequences depicting various polymorphic amino acid positions, enabling an efficient contact with the numerous bound peptides [31]. Three parameters, including IC_50_ value with <50 nM, the lowest percentile rank score, and high prediction score, were utilized to sort the best epitopes with the highest binding affinity for the chosen BoLA class-II molecule. This method allows the prediction of potential epitopes from characterized alleles with limited binding data.

The HTL epitopes further help to promote IFN-γ response to augment efficient innate and adaptive immunity. IFN-γ is considered to provide intrinsically safe responses by preventing virus replication [32]. Additionally, it provokes a multifaceted immune system by activating CTL and HTL against the virus. So, all HTL epitopes were subjected to IFN-γ inducing properties by utilizing the IFN-epitope server (http://crdd.osdd.net/raghava/ifnepitope/, accessed on 2 January 2021) [33] using SVM hybrid algorithms along with motif as per respective score [34,35].

#### 2.2.3. B-Cell Epitopes Prediction 

B-cell epitopes are essential components of the humoral immunity that elicits immunoglobulin (IgG) response. These linear epitopes are predicted using the ABCpred server (http://crdd.osdd.net/raghava/abcpred/, accessed on 20 February 2021) [36]. The server calculates the epitope prediction with high precision based on four parameters, such as specificity, sensitivity, accuracy, and positive predictive value. The input FASTA sequence of all proteins, with a set threshold at 0.5, was utilized to predict linear B-cell epitope that are unique, immunogenic, and continuous as output. The network generates a prediction precision with 65.93%, following several folds cross-validation.

### 2.3. Conservancy Analysis with All Global Isolates 

Epitope conservancy is the fraction of protein sequences containing epitopes at or above a certain identity level [37]. Each of the selected epitopes should account for conservancy as a consensus sequence among them. For this, each antigenic BEFV protein was aligned to generate a multiple sequence alignment (MSA) using the ClustalW module [38] of the MEGAX v.10.0.5 server [39]. Next, each predicted epitope was compared using the conservancy tool (http://tools.iedb.org/conservancy/, accessed on 1 February 2021) [37] with sequence identity at a set threshold of ≥80.

### 2.4. Multiepitope Vaccine Designing 

To construct a multi-epitope BEFV vaccine (MEV-BEFV), all selected epitopes of B cells, CTL, and HTL were connected with the aid of KK, AAY, and GPGPG linkers, respectively [20]. To enhance the immunogenicity, an adjuvant *β*-defensin (Uniprot ID—P46161) was added at the N-terminus [40] of the first BCE epitope via EAAAK linker [41]. However, a total of six different vaccine confirmations were developed by positioning predicted epitopes in different combinations.

### 2.5. Blast Analysis 

The final construct’s homology was inspected with the pBLAST server’s aid (https://blast.ncbi.nlm.nih.gov/Blast.cgi?PAGE=Proteins, accessed on 20 February 2021) against the bovine proteome by non-redundant (nr) collection of *Bos taurus* and *Bos indicus* proteins at NCBI, in order to cross-check for any similarities among these.

### 2.6. Physicochemical and Immunogenic Properties Assessment of the Vaccine Construct

The intention behind vaccination is to trigger an immune response when administered to the host. The candidate vaccine should, therefore, be stable, highly antigenic, non-allergic, and non-toxic in nature with good solubility. The physicochemical properties were evaluated using the ProtParam tool (https://web.expasy.org/protparam/, accessed on 19 February 2021) [42]. The various output parameters include amino acid (aa) composition, molecular weight, theoretical isoelectric point (pI), instability index (<40), estimated half-life in vitro and in vivo, stability profiling, aliphatic index, and the grand average of hydropathy [42].

Antigenicity prediction was done using the VaxiJen v2.0 tool owing to the highest precision (70–89%). Allergenicity screening was done by AllerTOP v2.0 (https://www.ddg-pharmfac.net/AllerTOP/, accessed on 30 January 2021) [43], at a set threshold of 0.4, and AllergenFP v1.0 (http://ddg-pharmfac.net/AllergenFP/, accessed on 20 February 2021), each providing accuracy of 88.7% and 87.9%, respectively. Toxicity discrimination and solubility tendency in *E. coli* upon overexpression were estimated using the ToxinPred server (http://crdd.osdd.net/raghava/toxinpred/, accessed on 19 February 2021) [44] and SOLpro (http://scratch.proteomics.ics.uci.edu, accessed on 19 February 2021) tool [45], respectively.

### 2.7. Secondary and Tertiary Structure Prediction 

The secondary structure of the MEV-BEFV peptide was predicted by CFSSP: Chou and Fasman secondary structure prediction server [46]. The server analyzes each amino acid’s relative frequencies in alpha helices, beta sheets, and turns, centered on known protein structures solved with X-ray crystallography. 

As the vaccine construct is a combination of many epitopes, the RaptorX server (http://raptorx.uchicago.edu/, accessed on 10 January 2021) [47] was utilized, as it works on deep learning modules and compares the input sequence with other non-redundant homologs. It allocates some confidence scores to the local and global model quality and a modeling error at each residue, suggesting a predicted 3D model [48]. The model was visualized by employing the Chimera tool [49]

### 2.8. Refinement, Model Quality Assessment, and Validation 

The refinement of the tertiary structure was done using 3D refine server (http://sysbio.rnet.missouri.edu/3Drefine/, accessed on 20 February 2021) [50]. It uses the CASP10 refining process to rebuild and repack the protein side-chains and utilizes force fields MD simulation to improve global and local structural quality to stabilize the overall protein model.

To verify the refined structure quality with that of the unrefined one, we utilized Ramachandran plot analysis using PROCHECK in SAVES v5.0 (http://servicesn.mbi.ucla.edu/PROCHECK/, accessed on 19 January 2021). The quality was evaluated by calculating the ϕ, ψ dihedral angles score of each aa residue (except for glycine and proline residue) in the energetically allowed and disallowed regions. Further, structural validation was done utilizing the ProSA-web server [51].

### 2.9. Bovine TLR7 Receptor and Vaccine Construct Molecular Docking 

An effective vaccine should adequately interact with the host’s immune receptors to generate an efficient immune response. The molecular docking computational approach is thus utilized to predict microscopic interactions between the interacting macromolecules. The literature survey revealed that NSSS RNA viruses are recognized by TLR7 immune receptor that generates the antiviral response. ClusPro 2.0 (https://cluspro.org/help.php, accessed on 2 February 2021) [52] and ZDOCK (http://zdock.umassmed.edu/, accessed on 2 February 2021) [53] servers were employed for the MEV construct (ligand) docking with TLR7 (receptor). The output produces several docked complex models with varying calculated electrostatic interaction values and the lowest Gibbs free energy scoring. Chimera tool was used to visualize all the docked complexes.

However, because of the unavailability of the crystal structure of *b*TLR7 in the protein data bank, the RaptorX server was utilized for model building, followed by refinement and its validation, as performed in Section 2.6 and Section 2.7.

### 2.10. Molecular Dynamic Simulation of the MEV-BEFV and the Vaccine bTLR7 

To evaluate the vaccine’s stability with TLR-7 and compare the vaccine protein’s dynamic behavior alone and its complex, molecular dynamic simulations were performed using the AMBER 18 package [54]. The AMBERff14SB forcefield [55] was used for assigning parameters to proteins. An appropriate amount of counterions was added to neutralize the system, and systems were solvated in an octahedron periodic box with the TIP3P water model [56]. Minimization was performed for each solvated system. Then, the system temperature and pressure were maintained at 300 K and 1 atm for 50 ps using the Langevin thermostat [57] and the Berendsen Barostat [58], respectively. The long-range electrostatic interactions were considered using the particle mesh Ewald (PME) scheme [59], and the non-bonded cut-off was taken as 10 Å. Lennard–Jones potential was used to treat the van der Waals interactions. In each case, MD simulation was run for 20 ns, and the protein-protein complex stability was obtained by root mean square deviation (RMSD) and root mean square fluctuation (RMSF) analysis from the trajectories.

### 2.11. Codon Adaptation and In Silico Cloning

Codon adaptation is a unique way to enhance the codon usage as per the host system for a higher protein expression rate. For example, *E. coli* K12 codon usage is different from the bovine host. Therefore, Java Adaptation Tool (JCat) (http://www.jcat.de/, accessed on 20 February 2021) [60] based on codon adaptation index values (CAI) was utilized. The generated peptide was reverse translated utilizing the EBI-EMBL server tool (https://www.ebi.ac.uk/Tools/st/emboss_backtranseq, accessed on 20 February 2021). Next, the optimized cDNA was subjected to the NEBcutter v2.0 (https://nc2.neb.com/NEBcutter2/, accessed on 20 February 2021) to check for XhoI and BamH1 restriction sites. The prokaryote ribosome binding and rho-independent terminator sites were also checked. Finally, the SnapGene tool (https://www.snapgene.com/snapgene-viewer/, accessed on 20 February 2021) was used to design this cDNA sequence into the pET28a (+) vector by adding to BamH1 and XhoI restriction site at the N- and C-terminal sites, respectively, to ensure the restriction cloning.

## 3. Results

### 3.1. Sequence Retrieval and Antigenicity Prediction 

All the examined structural proteins were depicted to be the primary antigenic targets for epitope prediction, as determined by antigenicity scores obtained by VaxiJen 2.0 tool (Edward Jenner Institute, Bulgaria, Europe) (Table 1). The L protein was not included in this study owing to its polymerase role in the replication process and few domains are similar to the host DNA polymerase. Eventually, utilizing various immunoinformatics tools, the above proteins were explored for epitope prediction.

### 3.2. Epitope Prediction of B-Cell and T-Cell Epitopes

Epitopes showing significant strong binding affinities with a common experimentally validated allele are excellent choices for MEV construct design. Unfortunately, for a bovine host, only a few BoLA alleles have been reported in relevance to pathological conditions [61]. Moreover, limited availabilities of predicting server tools are an additional constraint for predicting epitope binding to the BoLA class-I/II molecules. Thus, we extensively curated all individual alleles reported in the literature to predict the best probable epitopes with the highest binding affinity [62,63,64]. Further, epitopes binding to multiple alleles were considered the most suitable for their potent defense capabilities. 

#### 3.2.1. CTL Epitope Prediction

The CTL epitopes of 9–12 mer length based on the high binding affinity (9–12 mer length) with various BoLA class-I alleles were first predicted. Based on a high prediction score and the lowest percentile rank obtained against the BoLA molecules, the corresponding epitopes were marked as “strong interactions”. A total of 19 BoLA alleles were proposed (Table 2) against the long list of alleles (Appendix A), with the majority showing high affinity to N protein followed by G and P protein, having 17, 7, and 1 alleles, respectively, with none for the M protein (as it could not optimally fulfil the scoring criteria). Interestingly, a few of the strongly interacting alleles such as BoLA-2:00501, BoLA-2:00601, BoLA-2:00602, BoLA-2:00801, BoLA-2:00802, and BoLA-2:01201 showed commonality with N and G proteins, inferring their strong immune potential against the pathogen.

#### 3.2.2. HTL Epitope Prediction 

In this experiment, 15 mer HTL epitopes were selected based on the IC_50_ value of ≤50 nM, lowest percentile rank score, and highest prediction score, depicting high epitope binding affinity with the different subtypes of BoLA *DRB3* allele. Previous reports suggested this allele’s predominant involvement in pathogen presentation to the immune system [61]. Additionally, it represents the most polymorphic bovine MHC gene with 136 numerous alleles reported to date [65], and hence was specifically selected for HTL recognition (Appendix A, includes the list of all alleles explored during the study). Notably, the G protein displayed the best single epitope, as it strongly interacted with five subtypes of the BoLA DRB3 allele. Of the total seven allele subtypes, BoLA-*DRB3**0101 showed the highest defense potential, as these were encountered by all four studied proteins, followed by BoLA-DRB3*0303, which was seen in all, except P protein (Table 3). Surprisingly, the M protein achieved significant HTL epitope prediction results, although none of the epitope could signify a potential CTL epitope in this prediction.

On further mining for cytokine production among the selected HTL epitope, all showed IFN-γ induction characteristics and were considered propitious epitopes for vaccine design.

#### 3.2.3. B-Cell Epitopes Prediction 

The ABCPred server predicted B-cell epitopes from all the targeted proteins. Four epitopes from each of the viral proteins were scrutinized based on the highest-ranking among all the predicted epitopes, relating to the strong binding affinity to the B-cell receptors. These epitopes were selected as potent B-cell epitopes for vaccine construction (Table 4).

### 3.3. Conservancy Analysis

The multiple sequence alignment for each of the above-selected epitopes revealed a high percent identity among all global BEFV strains (Figure 2). Further, these epitopes showed a conservancy pattern at >80%. Based on the sequence alignment results, N protein showed 100% identity for all epitopes followed by G, M, and P protein with varying identity values (Appendix A). Thus, the resultant design indicates a broader efficacy against globally prevalent BEFV strains.

### 3.4. Multiepitope Vaccine Construction 

All the prioritized epitopes with the potential to induce humoral as well as a CMI response were adjoined by linkers and an adjuvant was included for MEV-BEFV design. The adjuvant (β-defensin) and consecutive 4 B-cell epitopes, 23 CTL, and 7 HTL epitopes were connected by EAAAK, AAY, GGGGS, and KK linkers, respectively, to form a single construct, maintaining its overall reactivity. The final vaccine constructs resulted in having 536 aa residues (Figure 3). Additionally, each of the predicted epitopes was variously configured to generate six different orientations to obtain the best configuration model.

### 3.5. Physicochemical and Immunogenic Properties Assessment of the MEV-BEFV 

Initially, by performing Blast-p analysis, we ascertained the non-homology of the constructed vaccine for the bovine host. Moreover, the vaccine construct showed an antigenic score of 0.52 at a 0.4% threshold with a VaxiJen server and was found to be non-allergenic, non-toxic, and highly soluble. We next performed physicochemical analysis. The molecular weight of the construct was calculated to be 58.72 kDa, reflecting good antigenic nature and ability for easy purification. The proteins below 110 kDa can be easily purified and thus are the preferred choice for large-scale production. The pI value of 9.32 indicates the basic nature of the peptide. The extinction coefficient’s value was 90,650 at 0.1% absorption, which assumed all cysteine residues are reduced. The protein’s half-life was estimated be >30 h in mammalian reticulocytes; >20 h in yeast; and >10 h in *E. coli*, suggesting its ability for prolonged exposure and stimulation of the host’s immune system. Further, an instability index of 34.97 confirmed the construct’s stability. The aliphatic index of 86.86 and GRAVY (grand average of hydropathy) index of −0.047 revealed high thermostability and hydrophilicity characteristics, respectively, suggestive of improved interactions within the body’s polar environment. Taken together, all results suggested the construct to be a suitable potential vaccine candidate.

### 3.6. Structure Prediction of MEV-BEFV

The secondary structure of the antigenic peptide determined the propensity of a sequence to form α-helix, β-sheets, and loops/turns, representing 70.5%, 31.3%, and 13.6% each type, respectively (Appendix A). These probably demonstrate the intra-and inter-chain interactions that could be involved in protein fooling in the actual environment.

The 3D modeling of the same construct was carried out using the RaptorX server. The best model with id-4d6wA was selected as per the calculated *p*-value of 4.47 × 10^−7^. The *p*-value reflects the model quality, and a value <10^−3^ presents higher model quality. In this experiment, 100% sequence residues were modeled for the 3D structure.

### 3.7. Structural Evaluations of Vaccine Construct

Further structural stability enhancement was achieved by refinement and validation of MEV residues. Of all the models generated by 3D refinement, model 5 represented the best-refined model, centered on relative scores for various parameters and ordered according to the RWPlus score. This determines the potential energy of the refined model. A lower score generally indicates a better-quality model. The RWplus score for the best refined model was found to be −81,905.9. Accordingly, we selected model 5, for which the obtained 3D refinement score is also low, with a value of 38,774.2, showing a model with better quality. Additionally, the obtained Global Distance Test-total score (GDT-TS) score and Global Distance Test-High Accuracy (GDT-HA) score were observed to be 0.99 and 0.97, respectively. The values of GDT-TS and GDT-HA varied in the range of [0, 1]; a similar score with respect to the initial model. A higher score indicates conservative refinement. The obtained RMSD from 3D-refinement was 0.343 Å; this is the CA RMS deviation score of the refined model with respect to the initial model. Based on these results, we can conclude that the predicted model qualifies as the better-quality model and can be used for further analysis (Figure 4A).

In quality check analysis by ERRAT, the refined model score was 50.77 and the Z score by ProSA was −5.39 (Figure 4B). Further validation depicted 86.2% (394) residues in the favored region with 12.9% (55) and 0.9% (4) residues in the allowed region and outlier region, respectively, by Ramachandran plot analysis (Figure 4C). These results inferred the refined model to be of good quality.

Additionally, the flexibility of the refined MEV model was examined by CABS-flex 2.0 server with default parameters. All the retrieved models exhibited a high fluctuation near the *N*-terminal compared with other regions, as displayed by the top finalized model (Figure 5A). The resultant contact map also displayed a well-defined residue–residue interaction pattern (Figure 5B). Further, the root mean square fluctuation (RMSF) plot was depicted and is shown in Figure 5C. It shows that higher fluctuating residues were seen between residue 401 and 451, with a maximum fluctuation of ~6 Å. Its N-terminal region exhibited higher fluctuations than other portions; this may be due to predicted loop regions, and it is not involved in the interaction with TLR-7.

### 3.8. Molecular Docking of the MEV-BEFV with bTLR7

By conducting a molecular docking study, the assessment of the binding affinity of the refined vaccine construct with the *b*TLR7 immune receptor was done to evaluate its appropriate immune activation (Figure 6). Using the ClusPro server, 30 standard outputs were obtained. Among these, the model with the lowest binding energy score was selected as it represents good binding affinity. As model 1 has the lowest energy criterion of −1551.9, it was selected as the best-docked complex. Besides, the ZDock server was utilized to reconfirm the above docking outputs that displayed the results as a Z-score value of 2384.11 for the docked complex. These results indicated that this MEV is the best suitable candidate for the vaccine. The best-fit model was finally chosen for the molecular dynamics simulation study among all the received models as an output.

### 3.9. Molecular Dynamics Simulation of the Docked-Complex

To further evaluate the stability of the docked protein with the TLR-7 complex, molecular dynamics (MD) simulations were performed using the AMBER package. The structural stability of the docked complex was evaluated by calculating the RMSD, as shown in Figure 7A. This indicates that the average deviations of apo vaccine protein and its docked complex with TLR-7 were 13 Å, and 11 Å, respectively. This reveals that the vaccine construct is stabilized upon interaction with TLR-7. It depicts that, in both systems, its initial deviation was increased till 10–12 ns, which later on becomes steady and stabilized after 15 ns. Similarly, the RMSF plot of the vaccine protein was compared with the docked complex and is shown in Figure 7B. It reflects that each of the residual fluctuations of the vaccine construct in its apo state were quite high as compared with its complex with TLR-7, suggesting that, upon interaction with TLR-7, the protein is stabilized. Primarily, the fluctuation was lowered at the C-terminal portion, ranging from 250 to 536 residues, indicating the interacting portion of the vaccine. Further, both the temperature and pressure were equilibrated for systems at 300 K and 1 atm pressure, respectively (Appendix A). Thus, MD results reveal that the docked complex of vaccine–bTLR-7 is stabilized.

### 3.10. In Silico Cloning of Vaccine Construct 

The host expression system in *E. coli* K12 strains differs and presents requirements for codon adaptation. This codon usage is usually denoted by the codon adaptation index (CAI) of the optimized sequence. By utilizing JCAT, the resultant adapted cDNA with a CAI value of 0.99 and 49.69% GC content indicated good expression probability in bacterial strain K12. The CAI of >0.8 and GC content of 30–70% are desirable for a high expression level [66]. Furthermore, this optimized sequence was reversed to ensure complementation in the direction of vector translation, and restriction sites BamH I and Xho I were added to 5′ and 3′ end, respectively. The pET28a (+) expression vector was utilized to insert this sequence using SnapGene software. Finally, with a sequence length of 6943 base pairs, a recombinant plasmid was generated that holds the ability for successful expression in the *E. coli* system (Figure 8).

## 4. Discussion

BEFV is a re-emerging livestock virus with enormous economic impacts in the dairy sector. The rapid rise in BEFV outbreaks necessitates the development of accelerated vaccination programs for its prevention and control [67]. Vaccines play a vital role in order to stimulate robust immune responses and defend against certain infectious diseases. Conventional vaccine development procedures are laborious and costly. However, the involvement of immunoinformatics tools provides various in silico and databases resources that could accurately be utilized to generate a simple, fast, specific, and reliable vaccine-designing approach. With the technological advances, the possibility to predict accurate immunogenic epitopes derived from an infectious agent has paved the way to design and develop effective multiepitope vaccines [68,69]. Interestingly, such vaccines are advantageous over the monovalent vaccine owing to their inherent potency to cumulatively elicit innate, humoral, and cellular immune responses [70].

Several multiepitope vaccines have been reported previously that employed immunoinformatics approaches with promising results against SARS-CoV-2 [71], Chandipura virus [72], HIV [73], Nipah virus [74], Ebola virus [75], Zika virus [76], and so on. In this study, a multiepitope vaccine construct containing B-cell, CTL, and HTL epitopes linked to an adjuvant and linkers was designed that could effectively provoke the host’s innate and adaptive immune responses, proposing it to be a strong candidate for the BEFV vaccine development approach.

For efficient MEV design, we specifically selected BEFV structural proteins (N, P, M, and G) owing to their regulatory roles in virus infectivity and pathogenicity, excluding L protein, as it majorly contributes to the viral replication process. All the structural protein sequences of Indian BEFV isolate as retrieved from NCBI were analysed for their antigenic potential and were subsequently exploited to predict B- and T-cell epitopes, with IFN-γ particularly inducing the ability of HTL epitopes. The B-cell elicits a humoral immune response that leads to virus neutralization and memory to defend any repeated exposure, but it often imparts low effectiveness and weakens over time [77,78]. Alternatively, the CMI response induced by CTL and HTL precisely limits pathogen spread either by killing infected cells or by secreting antiviral cytokines that enable life-long immunity [79,80,81]. Hence, each epitope type was included in vaccine construction.

Generally, epitopes with strong binding affinity with experimentally validated alleles represent an excellent choice to be included in the design of MEV construct [82]. Although a previous study suggested B-cell epitope prediction utilizing only the BEFV G protein [83], in contrast, our work targeted the entire set of structural proteins for such predictions. Our results generated linear B-cell epitopes with the highest score that were selected for each protein. For CTL and HTL epitope prediction, a comprehensive investigation of all susceptible BoLA alleles of class I/II molecules was conducted. Owing to the limited literature available on BoLA alleles, we extensively explored all individual alleles for epitope prediction scrutiny in the context of different pathologies. Among the numerous BoLA-I molecules, a few alleles are linked to an operative antigenic peptide with high affinity, while for the BoLA-II molecule, a single BoLA DRB3 allele established for pathogen recognition showed high affinity for various epitopes. Thus, the highest-ranked epitopes (CTL and HTL), as obtained at a very conservative threshold as recognized by BoLA class-I/II, were chosen for construct proceedings. Moreover, all HTL epitopes were examined for IFN-γ induction, which will promote the activation, development, and differentiation of immune cells such as B- and T-cells, macrophages, and so on. [84]. Our study targeted the bovine genome for predicting the most probable HTL epitopes, in contrast to other studies that utilized HLA for such prediction [85]. Nevertheless, conservancy assessment of these demonstrated high identity among the aa residues, implying the vaccine’s broad effectiveness against all global BEFV isolates rather than confining it to the local isolates. Following confirmation against all criteria, the prioritized epitopes were chosen for vaccine construction using linkers and adjuvants. The N-terminus of the MEV construct was affixed with an adjuvant, and epitopes were connected using KK, AAY, and GPGPG linkers. Linkers are incorporated as an essential element in the vaccine construct as they enhance the expression, folding, and stability of the independent domains [86]. Adjuvant β-defensins, a TLR3 agonist, acts as a strong immunostimulant that provokes an immune response by binding to its corresponding receptors, TLRs and CCR6, activating both immature dendritic cells and naïve T-cells at the infection location [40]. To enhance efficacy, stability, and sustainability, an adjuvant is implemented as an integral feature in vaccine design [87,88].

The vaccination’s prime objective is to induce a rapid immune response with limited adverse effects on the host’s body. Therefore, the designed construct was validated against the bovine proteome, which displayed no similarity, and ascertained its safety inside the host. Subsequently, highly stringent selection criteria profoundly suggested the construct to be non-allergic, highly antigenic, and non-toxic with highly soluble characteristics, along with optimum physiochemical properties. A comprehensive structural evaluation utilizing 3D refinement and Ramachandran plot illustrated a stable and good quality model with a near-native structure.

Previous reports on NSSS RNA viruses have shown that innate immune response activation mainly occurs through TLR7 and TLR8 receptors that reside on the immune cells’ surface. The up-regulation of TLR7 has been clearly shown for vesicular stomatitis virus infection [89], a close relative of BEFV. Hence, the designed vaccine was evaluated for its association with *b*TLR7 by performing molecular docking analysis. The docking score showed a substantially high binding affinity with stabilized interaction between the ligand–receptor complex, further confirmed by molecular dynamics simulations. It thereby indicated that the generated vaccine would induce TLR activation to contribute to stronger immune responses within the host.

Gene expression will differ across different hosts because of the mRNA codon’s inconsistency, so codon optimization is critical for higher expression [73]. The appropriate CAI value and GC content of the codon-optimized vaccine sequence suggested a higher expression in the *E. coli* K12 strain. As earlier studies suggested, *E. coli* is the most preferred and recommended system for bulk production of recombinant proteins [90]. Using the pET28(+) vector with the His-tagged vaccine sequence, in silico restriction cloning was eventually carried out to be easily purified to synthesize potential candidate vaccines on a larger scale [91]. Collectively, our integrated immunoinformatics approach would reinforce the therapeutic findings and improve the BEFV prophylaxis.

## 5. Conclusions

The high burden of the BEFV population often alarms the enormous economic concerns associated with and urges prophylaxis in the BEFV endemic countries for pathogen elimination. Indeed, an efficient vaccine’s effect is its ability to provide life-long immunity, protecting against repeated infection episodes. This study was undertaken to design a multi-epitope subunit vaccine for BEFV by utilizing an integrated immunoinformatics approach. The proposed MEV-BEFV model exhibited a number of characteristics that could theoretically stimulate cellular and humoral immune responses and will potentially contribute to BEFV vaccine development. While maintaining a high conservancy, the construct optimally complies with the required standards of antigenicity, allergenicity, toxicity, and various physicochemical parameters. The molecular docking and dynamic simulation revealed a significantly strong binding affinity to *b*TLR7, implying high stability at the physiological pH range. Additionally, to ensure good expression within the widely used *E. coli* K12 strain, thorough codon optimization and in silico restriction cloning were performed. Our findings, however, are the sole outcome of a computer-aided technology that paved the way for future in vivo and in vitro assessments to confirm the vaccine’s reliability, effectiveness, and safety of the vaccine constructs.

## Figures and Tables

**Figure 1 vaccines-09-00925-f001:**
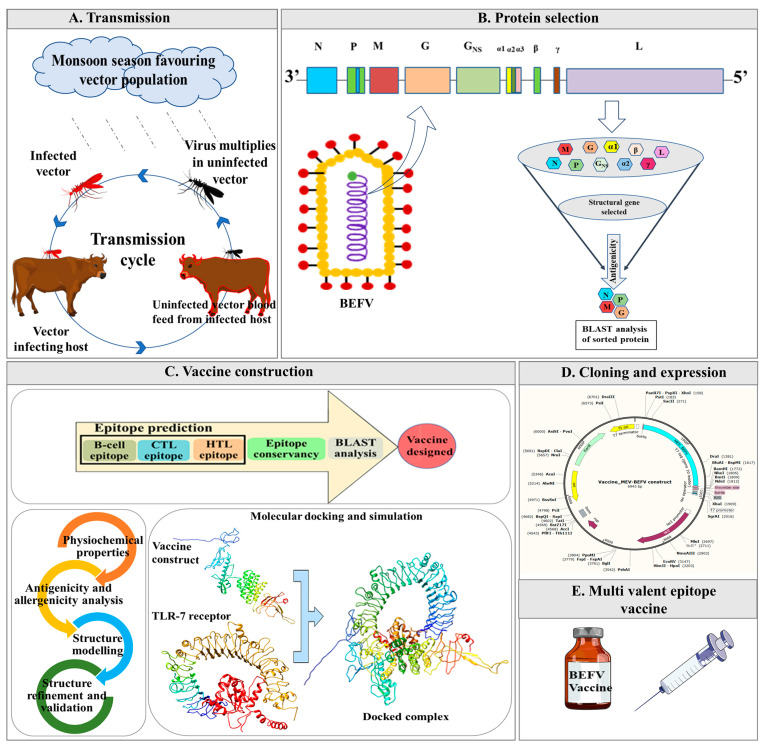
Schematic representation of the designed workflow involved in multi-epitope vaccine construction against bovine ephemeral fever virus (MEV-BEFV) using the immunoinformatics approach. It summarizes the steps involved during the process, starting from the (**A**) Viral disease transmission. (**B**) Antigenic viral protein selection, followed by (**C**) Vaccine construction that includes potent antigenic epitopes prediction. Subsequently, 3D modeling accompanied by a comprehensive assessment of different parameters of the assembled construct. Molecular docking and simulation with *b*TLR7 (Toll-like receptor 7) were then pursued to verify their affinity and interaction with the bovine host. Next, (**D**) In silico cloning compatible with prokaryotic expression system was evaluated for large-scale antigen production. Finally, (**E**) an Multivalent epitope vaccine (MEV) that is efficient, safe, and broadly effective against all global BEFV isolates was designed as an outcome. CTL, cytotoxic T-lymphocyte; HTL, helper T-lymphocyte.

**Figure 2 vaccines-09-00925-f002:**
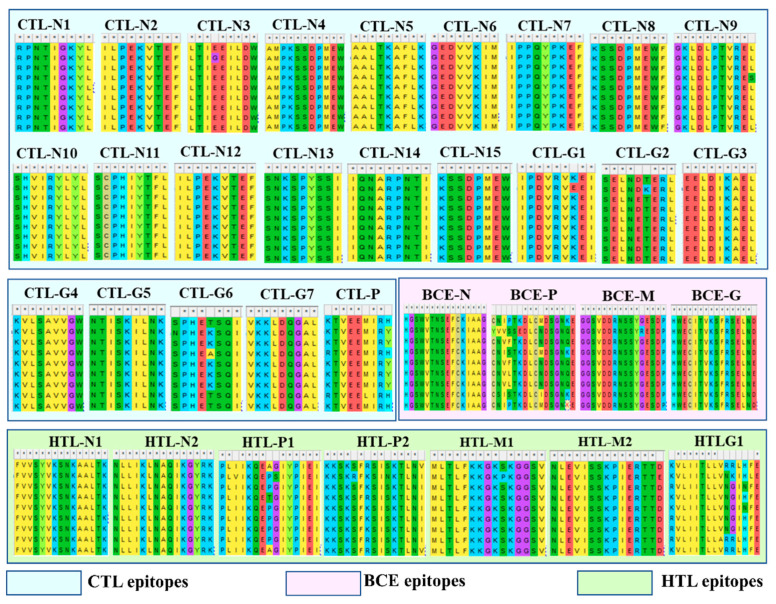
Conservancy analysis among the globally available BEFV isolates of all chosen predicted epitopes. The consensus sequence indicates the conservancy level for each epitope. Multiple sequence alignment revealed high sequence identity at a set threshold of ≥80 for all the predicted epitopes compared with the global BEFV isolates. The amino acid residues are represented as per their universal and color codes of the grids displaying the amino acid in reference to the MEGA-X server. The colored boxes represent the particular predicted epitope type as, HTL- and CTL -epitopes and B-cell epitopes (BCE).

**Figure 3 vaccines-09-00925-f003:**
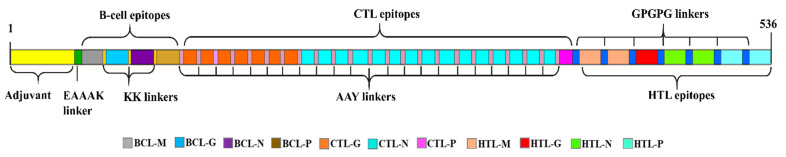
Graphical illustration of the multi-epitope vaccine constructs against BEFV. A 536 amino acid long vaccine construct consisting of an adjuvant (yellow) at the N-terminal end is linked with the multiepitope sequence through the EAAAK linker (green). The BCE, CTL, and HTL epitopes are fused with the help of KK (dark yellow), AAY (purple), and GPGPG (blue) linkers, respectively.

**Figure 4 vaccines-09-00925-f004:**
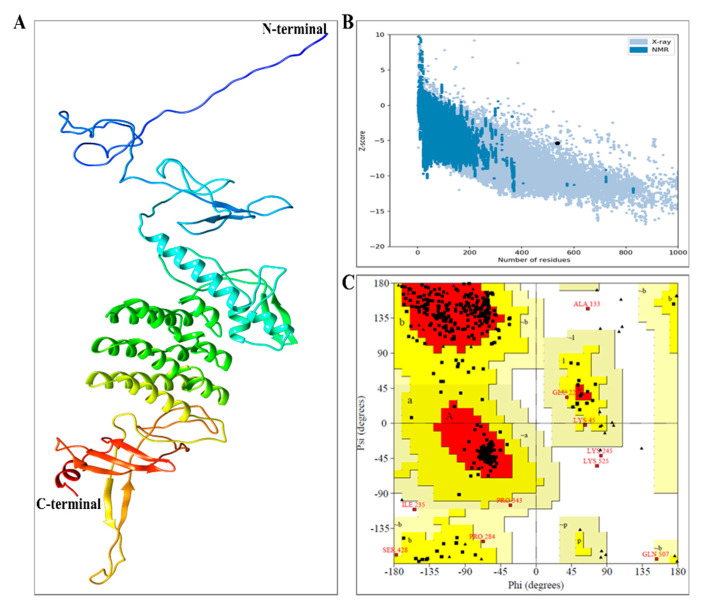
Illustrating the 3D model structural refinement, quality assessment, and validation of the vaccine construct. (**A**) Tertiary structure of the refined construct indicating α-helix, β-strand, and random coil. (**B**) ProsA Z-score (−5.39) indicates model quality closer to protein structure determined by X-ray crystallography, and hence the positively refined (**C**) Ramachandran plot of the refined model.

**Figure 5 vaccines-09-00925-f005:**
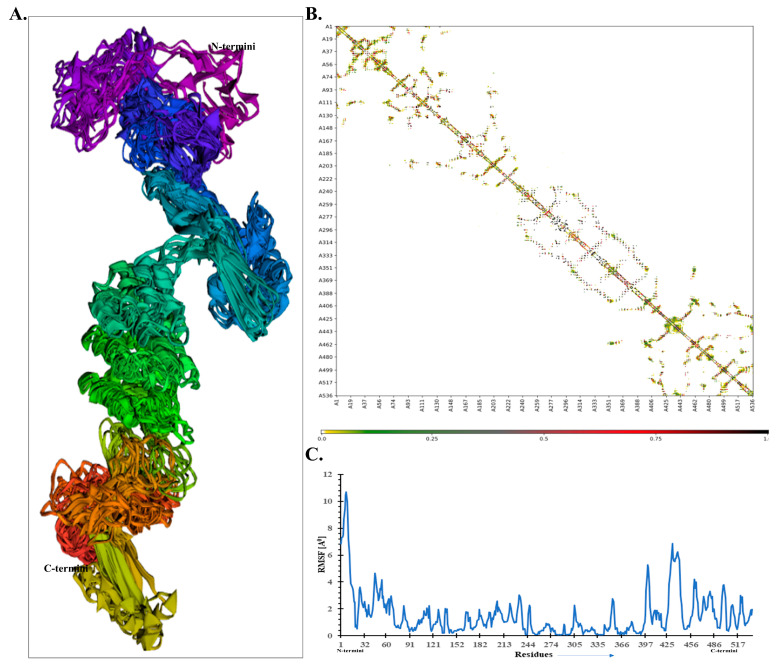
The MEV structural flexibility analysis. (**A**) Top 10 finalized models displaying obvious fluctuation. (**B**) Contact map comparing MEV residue–residue interaction. (**C**) Root mean square fluctuation (RMSF) plot suggesting the obvious fluctuations as observed in MEV residues during the simulation process.

**Figure 6 vaccines-09-00925-f006:**
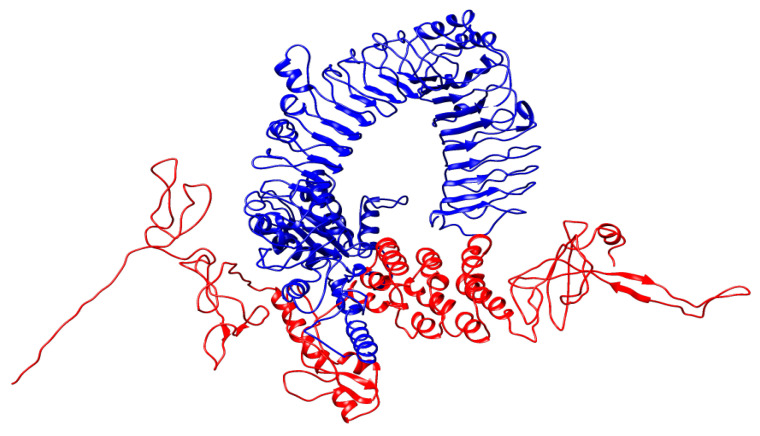
The docked complex of MEV-BEFV (ligand) and *b*TLR7 (receptor). A 536 amino acid long vaccine construct consisting of an adjuvant (yellow) at the N-terminal end is linked with the multiepitope sequence through the EAAAK linker (green). BCE, CTL, and HTL epitopes are fused with the help of KK (dark yellow), AAY (purple), and GPGPG (blue) linkers, respectively.

**Figure 7 vaccines-09-00925-f007:**
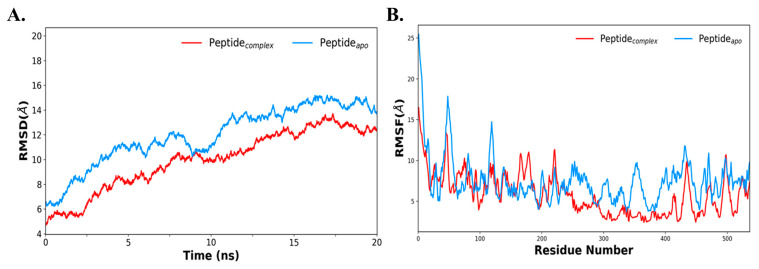
Illustrating molecular dynamics simulation at 20 ns. (**A**) RMSD plot of the peptide-apo (construct) and peptide complex (construct and *b*TLR7); (**B**) RMSF plot of the peptide-apo and *b*TLR7–MEV complex (red color depicts peptide complex with *b*TLR-7 and peptide-apo in blue).

**Figure 8 vaccines-09-00925-f008:**
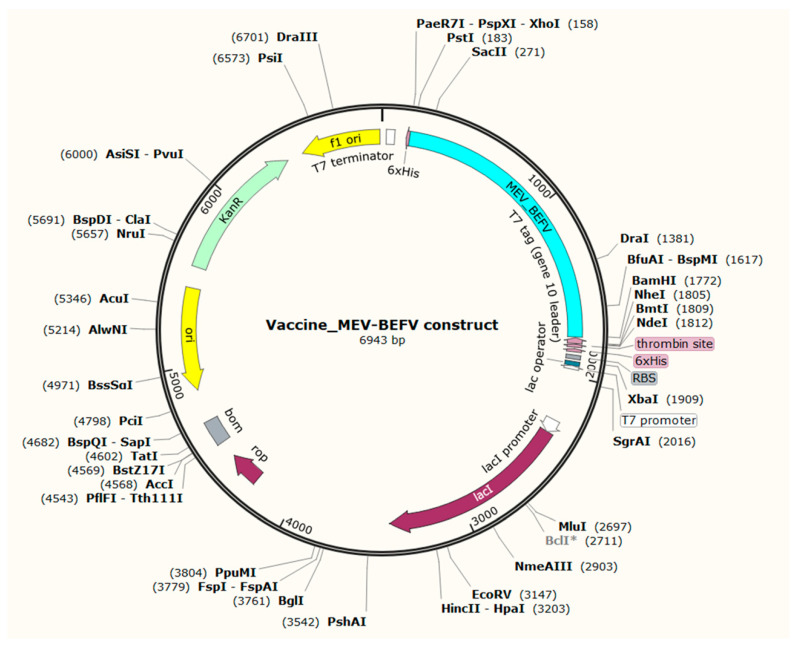
In silico cloning of the MEV-BEFV construct into pET28a (+) expression vector. The vaccine construct sequence was codon-optimized and inserted into the vector at the BamH1 and XhoI restriction sites. The light-blue-colored part represents the vaccine construct, while the remaining portion represents the vector backbone.

**Table 1 vaccines-09-00925-t001:** List of bovine ephemeral fever virus (BEFV) antigenic proteins utilized to design a multi-epitope vaccine for BEFVs and their corresponding VaxiJen value.

S. No.	Protein Name	Length (aa)	NCBI Protein ID	VaxiJen Score
1	Nucleocapsid protein	432	QOU09200	0.5222
2	Phosphoprotein	279	QOU09201	0.5095
3	Matrix protein	224	QOU09203	0.7226
4	Glycoprotein	624	QOU09204	0.4723

Set threshold for VaxiJen server is 0.4 for virus.

**Table 2 vaccines-09-00925-t002:** Predicted cytotoxic T-lymphocyte (CTL) epitopes of selected BEFV antigenic proteins and their interacting BoLA class-I alleles with their binding details. BoLA, bovine leukocyte antigen.

Protein	Selected Epitopes	BoLA Binding Alleles	Position	Prediction Score	%Rank
	RPNTIGKYL	BoLA-2:00501	419–427	0.492	0.04
		BoLA-2:00601		0.303	0.153
	ILPEKVTEF	BoLA-2:00602	399–407	0.260	0.097
	LTIEEILDW	BoLA-2:00801	238–246	0.740	0.053
	AMPKSSDPMEW	BoLA-2:00802	379–389	0.287	0.224
	AALTKAFLK	BoLA-2:01201	340–348	0.942	0.009
	GEDVVKIM	BoLA-2:01601	253–260	0.210	0.125
Nucleoprotein	IPPQYPKEF	BoLA-2:01602	19–27	0.330	0.019
	KSSDPMEWF	BoLA-2:04401	382–390	0.470	0.166
	GKLDLPTVREL	BoLA-2:02603	43–53	0.431	0.34
	SHVIRYLYL	BoLA-3:00102	66–74	0.087	0.203
	ILPEKVTEF	BoLA-3:00103	399–407	0.231	0.036
	SCPHIYTFL	BoLA-3:00201	291–299	0.693	0.044
		BoLA-3:00101		0.149	0.118
	SNKSPYSSI	BoLA-3:00401	282–290	0.448	0.009
	IQNARPNTI	BoLA-3:01101	415–423	0.627	0.071
	KSSDPMEW	BoLA-4:02402	382–389	0.605	0.2
Phosphoprotein	KTVEEMIRH	BoLA-1:00901	247–255	0.812	0.08
	IPDVRVKEI	BoLA-2:00501	192–200	0.308	0.199
	SELNDTERL	BoLA-2:00601	219–227	0.217	0.1
	EELDIKAEL	BoLA-2:00602	296–303	0.259	0.099
Glycoprotein	KVLSAVVGW	BoLA-2:00801	521–529	0.801	0.031
	NTISKILNK	BoLA-2:01201	310–318	0.902	0.022
	SPHETSQI	BoLA-2:01802	499–506	0.947	0.021
	VKKLDQGAL	BoLA-2:02602	116–124	0.684	0.057

Strong binders are defined as having % rank < 0.5.

**Table 3 vaccines-09-00925-t003:** Predicted helper T-lymphocyte (HTL) epitopes of selected BEFV antigenic proteins and their interacting BoLA class-II alleles with their binding details. Strong binders are defined as having lowest % rank; IC_50_ value in nm.

Protein	Epitope	Position	Allele	Predicted Score	IC_50_ Value	% Rank
	FVVSYVKSNKAALTK	330–344	BoLA-DRB3*0101	0.851	4.19	0.7
Nucleoprotein			BoLA-DRB3*0901	0.807	6.38	0.15
	NLLIKLNAQIKGYRK	154–168	BoLA-DRB3*0303	0.909	2.39	0.5
Phosphoprotein	PLIIKQEAGIYPIEI	52–66	BoLA-DRB3*6101	0.779	8.36	7
	KKSKSFRSISKTLNV	258–272	BoLA-DRB3*0101	0.776	8.58	3
Matrix protein	MLTLFKKGKSKGGSV	1–15	BoLA-DRB3*0101	0.658	26.79	5
	NLEVISSKPIERTTD	63–77	BoLA-DRB3*0303	0.813	6.02	2
	KVLIITLLVRRLHFE	3–17	BoLA-DRB3*0101	0.836	4.85	1
			BoLA-DRB3*2004	0.89	2.87	0.05
Glycoprotein			BoLA-DRB3*03021	0.865	3.66	2
			BoLA-DRB3*0303	0.851	4.2	3
			BoLA-DRB3*1101	0.926	2.03	0.05

**Table 4 vaccines-09-00925-t004:** Predicted B-cell epitopes of selected BEFV antigenic proteins with their binding scores.

Protein	Predicted Epitope	Position	Score
**Nucleoprotein**	HGSWVTNSEFCKIAAG	180–195	0.93
**Phosphoprotein**	CNIPTKDLCMDSGNKE	112–127	0.93
**Matrix protein**	GGSVDDRNSSYGESDP	12–27	0.94
**Glycoprotein**	HWECITVKSFRSELND	208–227	0.96

For each protein, of numerous predicted epitopes, only the top scoring epitopes were considered as a selection criterion.

## Data Availability

The data supporting the findings of this study are openly available in the GenBank repository (https://www.ncbi.nlm.nih.gov/genbank, accessed on 9 December 2020).

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
