# Peer review of "Immunoinformatics Approach to Design Multi-Epitope- Subunit Vaccine against Bovine Ephemeral Fever Disease"

_vaccines, 2021, doi:10.3390/vaccines9080925_

Round 1

Reviewer 1 Report

This manuscript entitled “Immunoinformatics approaches to explore Bovine Ephemeral Fever Virus (BEFV) structural proteome to develop a Multi-epitope based vaccine” by Pyashi et al. is a scientific piece of work. The whole work is based on the immunoinformatics approaches and performed well in a sequential manner. However, there are few limitations of this work that needs to be corrected before publication. The comments are mentioned below.

  1. The title should be modified as “Immunoinformatics approaches to explore Bovine Ephemeral Fever Virus (BEFV) structural proteins to design a Multi-epitope subunit vaccine” to make it more suitable.
  2. Abstract: This part is written well but needs to check the grammatical mistakes. Line 22: use sentence case for molecular dynamics word. Line 23: In silico should be in italic.
  3. Keywords: Keywords are mentioned correctly.
  4. Introduction:
  • Line 60-61: Needs referencing
  • Line 72: It should be corrected as “immunoinformatics”
  • Graphical abstract needs more clarity. It will be nice to increase the DPI of the same to make it clear to visualize.
  1. Methodology: This part is written well but needs a minor correction. Line 182: Authors should cite this sentence.
  2. Result
  • Figure 2: The conservancy analysis part is presented well, but its figure part needs to add the meaning of the different colors used in the figure and also the meaning of different letter codes used inside.
  • Table 1 and 3 is missing.
  • The HTL epitope screening part needs rewriting to explain the allele selection and epitope sorting criteria.
  • Figure 5b and 5c need more clarity in visual appearance
  1. Discussion and conclusion
  • This part is written well and does not need a correction

Author Response

Reviewer 1:

This manuscript entitled “Immunoinformatics approaches to explore Bovine Ephemeral Fever Virus (BEFV) structural proteome to develop a Multi-epitope based vaccine” by Pyashi et al. is a scientific piece of work. The whole work is based on the immunoinformatics approaches and performed well in a sequential manner. However, there are few limitations of this work that needs to be corrected before publication. The comments are mentioned below.

  1. The title should be modified as “Immunoinformatics approaches to explore Bovine Ephemeral Fever Virus (BEFV) structural proteins to design a Multi-epitope subunit vaccine” to make it more suitable.

Our response: We thank the reviewer for the support and critical analysis of the work. As suggested, we have reframed the title.

  1. Abstract: This part is written well but needs to check the grammatical mistakes. Line 22: use sentence case for molecular dynamics word. Line 23: In silico should be in italic.

Our response: We thank the reviewer for the support and for pointing out the errors. We regret the error, and we have made the necessary corrections.

  1. Keywords: Keywords are mentioned correctly.

Our response: We sincerely thank the reviewer for the critical analysis.

  1. Introduction:
  • Line 60-61: Needs referencing

Our response: We sincerely thank the reviewer for the critical analysis and have made the necessary citations as [15].

  • Line 72: It should be corrected as “immunoinformatics”

Our response: We regret the error and have made the necessary correction.

  • Graphical abstract needs more clarity. It will be nice to increase the DPI of the same to make it clear to visualize.

Our response: We believe the reviewer is pointing out the designed workflow (Fig 1), as there is a graphical abstract in the version. We now have increased the resolution to 600 dpi for better visualization.

  1. Methodology: This part is written well but needs a minor correction. Line 182: Authors should cite this sentence.

Our response: We thank the reviewer for the support and for pointing out the errors. We regret the error, and we have made the necessary citation as [41].

  1. Result
  • Figure 2: The conservancy analysis part is presented well, but its figure part needs to add the meaning of the different colors used in the figure and also the meaning of different letter codes used inside.

Our response: We sincerely thank the reviewer for the critical analysis and constructive suggestion. The color codes in the two-dimensional display grids indicate the amino acid it contains. The MEGA-X server utilizes the list of default colors based on the residues' biochemical properties (synonymous residues indicated by the same color). The color code is mentioned below the table.

Symbol

Color

Symbol

Color

A

Yellow

M

Yellow

C

Olive

N

Green

D

Aqua

P

Blue

E

Aqua

Q

Green

F

Yellow

R

Red

G

Fuchsia

S

Green

H

Teal

T

Green

I

Yellow

V

Yellow

K

Red

W

Green

L

Yellow

Y

Lime

Additionally, different letter codes used inside respresents the universal amino acid codes as per the International Union of Pure and Applied Chemistry and International Union of Biochemistry (IUPAC-IUB): Nomenclature and Symbolism for Amino Acids and Peptides (Recommendations 1983).

  • Table 1 and 3 is missing.

Our response: We regret the error. Some layout issues that happened during the layout and peer review stage led to missing Table 1 and Table 3. We have re-incorporated the missing Tables at our end as suggested.

  • The HTL epitope screening part needs rewriting to explain the allele selection and epitope sorting criteria.
  •  

Our response: We have reframed the sentences mention in the method and result section in the revised manuscript as suggested (Line no, methods: 126-133; result: 319-325). It follows as:

Method: The server is based on the advanced NN-align method that utilizes two steps to estimate the network weight configuration and peptide binding score (core). It works to provide the environment for peptide binding strength with the pseudo-MHC sequences depicting various polymorphic amino acid positions, enabling an efficient contact with the numerous bound peptides [30]. Three parameters that include IC50 value with <50 nM, the lowest percentile rank score, and lastly, high prediction score were utilized to sort the best epitopes with the highest binding affinity for the chosen BoLA class-II molecule.

In results: Selection of predicted HTL epitope of 15mer length was done based on the IC50 value of ≤50nM, lowest percentile rank score, and highest prediction score, depicting high epitope binding affinity with the different subtypes of BoLA DRB3 allele. Previous reports suggested predominant involvement of BoLA DRB3 allele in pathogen presentation to the immune system [55]. Also, being the most polymorphic bovine MHC gene with 136 numerous alleles reported to date [59], all included in the NetMHCIIpan 2.1 server was explicitly selected for HTL recognition.

  • Figure 5b and 5c need more clarity in visual appearance

Our response: We thank the reviewer for this suggestion. We have tried improving the transparency in visual appearance to the best possible.

  1. Discussion and conclusion
  • This part is written well and does not need a correction

                   Our response: We thank the reviewer for the support and critical analysis.

Reviewer 2 Report

With this article Shruti Pyasi and collaborators, attempt to select antigens for a multiepitope-based vaccine against prevalent variants of BEFV based on an immunoinformatic approach. They claim this selection would be enough to produce efficient protection. However, this goal is not met by the manuscript, but the authors are aware of the lack of evidence and they address it as an opportunity for “future laboratory experiments”. The manuscript is well-written, clear and contains detailed information on the basis of their selection. There are many instances where the authors fully explain their ideas and use bioinformatic evidence to support their model. In my opinion, their bioinformatic approach is a good start to select antigens. However, they do not have any real data, they do not create any new pipeline, rather than use webservers to copy and paste fasta sequences and get information that MIGHT or might not be real. Everything is a prediction, including their 20 ns molecular dynamic simulations or the claim that of the peptide half-life (line 344-347) or data not shown?. It would be interesting to confirm these predictions are real with experimental data.

Major issues:

Line 365-366.  “RMSD score of 0.343 â„« denotes aggressive refinement” or just very high entropy due to poor structural modeling.

Line 382-38. “the root mean square fluctuation (RMSF) plot for each amino acid in the MEV model ranged between 0.0 and 6.3 Å indicating high flexibility, affirming its efficient vaccine potentiality”. Definitely NO. What it shows is that they have a predicted a very structured confirmation in one part of the protein and a very non structured end. To determined it, usually people show RMSD changes by amino acid before a RMSF analysis.

Line 413-414. “Both these plots indicated the stable complex formation of vaccine peptides with TLR-7 receptors.” How to prove that a 7 â„« difference at the end of the simulation indicates stable complex formation if both peptides (complex and apo) have the same initial difference and through all the MD.

Line 528-529. “The designed vaccine showed various features to induce cellular and humoral immune responses with efficient memory in combating the pathogen”. I do not agree that this statement is supported by their results because the authors never tested it.

Minor issues:

Figure 4A. Please show where N- and C- terminals are located.  

Figure 5D. Should be supplementary. It really does not support any new evidence. The models does not have any labels, C- N- term or x y z axes.

Figure 7A. If I assume correctly, the graph should show 20 ns of MD. Please correct the numbers in the x-axis. A zero on the right after 2 was crop and it reads now 2 instead of 20.

Unfortunately, I cannot recommend this manuscript for its publication in Vaccines until (at least) the major issues are addressed.  

Author Response

Reviewer 2:

With this article Shruti Pyasi and collaborators, attempt to select antigens for a multiepitope-based vaccine against prevalent variants of BEFV based on an immunoinformatic approach. They claim this selection would be enough to produce efficient protection. However, this goal is not met by the manuscript, but the authors are aware of the lack of evidence and they address it as an opportunity for “future laboratory experiments”. The manuscript is well-written, clear and contains detailed information on the basis of their selection. There are many instances where the authors fully explain their ideas and use bioinformatic evidence to support their model. In my opinion, their bioinformatic approach is a good start to select antigens. However, they do not have any real data, they do not create any new pipeline, rather than use webservers to copy and paste fasta sequences and get information that MIGHT or might not be real. Everything is a prediction, including their 20 ns molecular dynamic simulations or the claim that of the peptide half-life (line 344-347) or data not shown?. It would be interesting to confirm these predictions are real with experimental data.

Our response: We sincerely thank the reviewer for the critical analysis and constructive suggestion. We regret the inappropriate sentence usage within the manuscript. The draft is based on a recently developed computational approach that proposes predictive data by integrating reverse vaccinology with immunoinformatics data. This approach presents a fast, easy, and affordable methodology without actually requiring the live pathogens against the conventional methods. It thus showcases the method of choice for probable vaccine development by researchers worldwide. While computationally aided designed vaccines significantly forecast their safety and precision, we explicitly state that additional lab trials should confirm their effectiveness and safety. We are aware of the fact that this is the starting point of our work. We regret that we couldn't provide any experimental data of our designed MEV-BEFV due to various reasons. We hope to get the different viral isolates prevalent globally, animal ethics approval, and arrange the necessary funding to carry out this work.   We, therefore, request to consider the current work as baseline publication for the future vaccine.

Major issues:

Line 365-366.  “RMSD score of 0.343 â„« denotes aggressive refinement” or just very high entropy due to poor structural modeling.

Our response: We thank the reviewer for the critical analysis. Now we reframed the sentences in the revised manuscript (Line no 397-408). It follows as:

The best model was taken into consideration and ordered based on the best RWPlus score. This determines the potential energy of the refined model. A lower score generally indicates a better quality model. The RWplus score for the best-refined model was found to be -81905.9. Accordingly, we have chosen model 5, for which the obtained 3D refine score is also low, with a value of 38774.2 that shows a better quality model. The obtained GDT-TS score and GDT-HA score were observed to be 0.99 and 0.97, respectively. GDT-TS and GDT-HA values vary in the range of [0,1]; it is the similarity score w.r.t initial model. A higher score indicates conservative refinement. The obtained RMSD from 3D-refinement was 0.343 Å; this is the CA RMS deviation score of the refined model w.r.t initial model. Overall, it can be concluded that the predicted model qualifies the better-quality model and can be used for further analysis.

Line 382-38. “the root mean square fluctuation (RMSF) plot for each amino acid in the MEV model ranged between 0.0 and 6.3 Å indicating high flexibility, affirming its efficient vaccine potentiality". Definitely NO. What it shows is that they have a predicted very structured confirmation in one part of the protein and a very non-structured end. To determined it, usually people show RMSD changes by amino acid before a RMSF analysis.

Our response: We sincerely thank the reviewer for this analysis. As pointed out, we have rectified the wordings and have reframed the sentences in the revised manuscript (Line no 420-428). It follows as:

Additionally, the flexibility of the refined MEV model was examined by CABS-flex 2.0 server with default parameters. All the retrieved models exhibited a high fluctuation near N-terminal compared to other regions, as displayed by the top finalized model (Figure 5a). The resultant contact map also displayed a well-defined residue-residue interaction pattern (Figure 5b). Further, the root mean square fluctuation (RMSF) plot was depicted and shown in Figure 5c. It shows that higher fluctuating residues were seen between residue 401 and 451 with a maximum fluctuation of ~ 6 Å. Its N-terminal region exhibited higher fluctuations than other portions, this may be due to predicted loop regions, and it is not involved in the interaction with TLR.

Line 413-414. “Both these plots indicated the stable complex formation of vaccine peptides with TLR-7 receptors.” How to prove that a 7 â„« difference at the end of the simulation indicates stable complex formation if both peptides (complex and apo) have the same initial difference and through all the MD.

Our response: We thank the reviewer for this critical analysis. We have reframed the sentences in the revised manuscript (Line no, 446-460). It follows as:

Further to evaluate the docked vaccine protein's stability with the TLR-7 complex, molecular dynamics simulations were performed using the AMBER package. The structural stability of the docked complex was evaluated by calculating the RMSD as shown in Figure 7A. This indicates that the average deviations of apo vaccine protein and its docked complex with the bTLR-7 were 13 Å, and 11 Å, respectively. It reveals that the vaccine protein gets stabilized upon interaction with the bTLR-7. It depicts that in both systems, its initial deviation was increased till 10-12 ns which later on becomes steady and stabilized after 15 ns. Similarly, the RMSF plot of the vaccine protein was compared with the docked complex and shown in Figure 7B. It reflects that each residual fluctuations of the vaccine protein in its apostate were quite high compared to its complex with the TLR-7, suggesting that upon interaction with the TLR-7, the vaccine protein gets stabilized. Primarily the fluctuation was lowered at the C-terminal portion ranging from 250-536 residues, indicating the interacting portion of the vaccine. Thus, MD results reveal that the docked complex of vaccine-TLR-7 is stabilized.

Line 528-529. “The designed vaccine showed various features to induce cellular and humoral immune responses with efficient memory in combating the pathogen”. I do NOT agree that this statement is supported by their results because the authors never tested it!

Our response: We sincerely agree with the reviewer for this critical remark. Although, as per the in silico based methodology, our designed subunit vaccine in principle indicates its possible effectiveness to fight against the pathogen, however, the experimental verification will confirm its efficacy. Due to the unprecedented pandemic situation, access to the laboratory is hampered, and thus wet lab confirmation is not likely possible at this stage. However, we will plan the experimental proof as our next future plan. Therefore, we have reframed the revised manuscript sentences from confirmative statement to predictive statement (Line no,573-577).

This study was undertaken to design a multi-epitope subunit vaccine for BEFV by utilizing an integrated immunoinformatics approach. The proposed MEV-BEFV model exhibited several characteristics that could theoretically stimulate cellular and humoral immune responses and contribute to BEFV vaccine development.

Minor issues:

Figure 4A. Please show where N- and C- terminals are located.  

Our response: We regret the error. We now have clearly labeled the N and C-termini of peptide in Figure 4A.

Figure 5D. Should be supplementary. It really does not support any new evidence. The models does not have any labels, C- N- term or x y z axes.

Our response: Thank you for your suggestion, but since there is no figure named Fig 5D, we could not reset it as a supplementary file. Please let us know if there is a typo error. However, we have clearly labeled the N and C-termini of peptides in all the 10 models in Figure 5A.

Figure 7A. If I assume correctly, the graph should show 20 ns of MD. Please correct the numbers in the x-axis. A zero on the right after 2 was crop, and it reads now 2 instead of 20.

Our response: We sincerely regret the error. We now have corrected the number from 2 to 20 as it has been cropped by mistake.

Unfortunately, I cannot recommend this manuscript for its publication in Vaccines until (at least) the major issues are addressed.  

Our response: We tried to address all the issues (both major and minor) to the best of our level. We are hopeful that this draft now fulfills the required criteria for publication.

Round 2

Reviewer 2 Report

In this revised version of the manuscript, Shruti Pyasi and collaborators address all the issues of concern and I fully recommend this manuscript for its publication in Vaccines. Their computational findings will provide a good candidate for a more effective vaccine and a good example of a new area of vaccine development.